# Applications of Nanocellulose/Nanocarbon Composites: Focus on Biotechnology and Medicine

**DOI:** 10.3390/nano10020196

**Published:** 2020-01-23

**Authors:** Lucie Bacakova, Julia Pajorova, Maria Tomkova, Roman Matejka, Antonin Broz, Jana Stepanovska, Simon Prazak, Anne Skogberg, Sanna Siljander, Pasi Kallio

**Affiliations:** 1Department of Biomaterials and Tissue Engineering, Institute of Physiology of the Czech Academy of Sciences, Videnska 1083, 14220 Prague, Czech Republic; Julia.Pajorova@fgu.cas.cz (J.P.); Roman.Matejka@fgu.cas.cz (R.M.); Antonin.Broz@fgu.cas.cz (A.B.); Jana.Stepanovska@fgu.cas.cz (J.S.); Simon.Prazak@fgu.cas.cz (S.P.); 2Faculty of Biotechnology and Food Sciences, Slovak University of Agriculture in Nitra, Tr. A. Hlinku 2, 94976 Nitra, Slovakia; xtomkovam2@uniag.sk; 3BioMediTech Institute and Faculty of Medicine and Health Technology, Tampere University, Korkeakoulunkatu 3, 33014 Tampere, Finland; anne.skogberg@tuni.fi (A.S.); pasi.kallio@tuni.fi (P.K.); 4Automation Technology and Mechanical Engineering, Faculty of Engineering and Natural Sciences, Tampere University, Korkeakoulunkatu 6, 33720 Tampere, Finland; sanna.siljander@tuni.fi

**Keywords:** nanofibrillated cellulose, cellulose nanocrystals, fullerenes, graphene, carbon nanotubes, diamond nanoparticles, sensors, drug delivery, tissue engineering, wound dressing

## Abstract

Nanocellulose/nanocarbon composites are newly emerging smart hybrid materials containing cellulose nanoparticles, such as nanofibrils and nanocrystals, and carbon nanoparticles, such as “classical” carbon allotropes (fullerenes, graphene, nanotubes and nanodiamonds), or other carbon nanostructures (carbon nanofibers, carbon quantum dots, activated carbon and carbon black). The nanocellulose component acts as a dispersing agent and homogeneously distributes the carbon nanoparticles in an aqueous environment. Nanocellulose/nanocarbon composites can be prepared with many advantageous properties, such as high mechanical strength, flexibility, stretchability, tunable thermal and electrical conductivity, tunable optical transparency, photodynamic and photothermal activity, nanoporous character and high adsorption capacity. They are therefore promising for a wide range of industrial applications, such as energy generation, storage and conversion, water purification, food packaging, construction of fire retardants and shape memory devices. They also hold great promise for biomedical applications, such as radical scavenging, photodynamic and photothermal therapy of tumors and microbial infections, drug delivery, biosensorics, isolation of various biomolecules, electrical stimulation of damaged tissues (e.g., cardiac, neural), neural and bone tissue engineering, engineering of blood vessels and advanced wound dressing, e.g., with antimicrobial and antitumor activity. However, the potential cytotoxicity and immunogenicity of the composites and their components must also be taken into account.

## 1. Introduction

Nanocellulose/nanocarbon composites are hybrid materials containing cellulose and carbon nanoparticles. Integration of nanocarbon materials with nanocellulose provides functionality of nanocarbons, using an eco-friendly, low-cost, strong, dimension-stable, nonmelting, nontoxic and nonmetal matrix or carrier, which alone has versatile applications in industry, biotechnology and biomedicine (for a review, see [1,2]). In addition to its advantageous combination with nanocarbon materials, nanocellulose is an appealing material for biomedical applications due to its tunable chemical properties, nonanimal origin, and resemblance to biological molecules in dimension, chemistry and viscoelastic properties, etc. [3,4,5,6].

Cellulose nanomaterials include cellulose nanofibrils (CNFs) and cellulose nanocrystals (CNCs) [3]. CNFs are manufactured using either a bottom-up or a top-down approach. The bottom-up approach involves bacterial (*Gluconacetobacter*) biosynthesis to obtain bacterial cellulose (BC), while, in the top-down method, cellulosic biomass from plant fibers is disintegrated into smaller CNFs [7] that contain amorphous and crystalline regions [3]. The fibrillation of cellulose is achieved using mechanical forces, chemical treatments, enzymes or combinations of these. After fibrillation, the width of CNFs is typically between 3 and 100 nm, and the length can be several micrometers [8]. Separation of the crystalline parts from the amorphous regions of the fibers or fibrils to obtain CNCs typically requires acid hydrolysis, which destroys the amorphous regions [9]. Entangled CNFs are longer, while CNCs possess shorter needle- or rod-like morphology with a similar diameter and a more rigid molecule due to their higher crystallinity [3,9]. In general, the properties of nanocelluloses are variable and depend on their origin, type, processing, pretreatments and functionalization. Integration with other materials, as well as fabrication of the final product, further affects the properties of the resulting composite or hybrid structure.

Carbon nanoparticles include fullerenes (usually C_60_), graphene-based particles (graphene, graphene oxide, reduced graphene oxide, graphene quantum dots), nanotubes (single-walled, double-walled, few-walled or multi-walled) and nanodiamonds (for a review, see [10,11,12,13,14,15,16,17,18,19,20]). The most frequently used nanocellulose/nanocarbon composites contain graphene or carbon nanotubes, while composites of nanocellulose with nanodiamond, and particularly with fullerenes, are less frequently used. Other carbon nanostructures, which are less frequently used in nanocellulose/nanocarbon composites, at least for biomedical applications, include carbon nanofibers [21,22,23,24,25], carbon quantum dots [26,27,28] activated carbon [29,30] and carbon black [31,32,33].

Nanocellulose/nanocarbon composites can be prepared in one-dimensional (1D), two-dimensional (2D) or three-dimensional (3D) forms. 1D composites are represented, for example, by C_60_ fullerenes grafted onto cellulose nanocrystals that have undergone amination or oxidation [34,35]. 2D composites are represented by films, which can be self-standing or supported, i.e., in the form of free-standing membranes [29,36,37,38,39,40,41] or in the form of coatings deposited on bulk materials [33,42]. The films can be formed by depositing carbon nanoparticles on a nanocellulose layer [43,44]. More frequently, however, they are fabricated from aqueous dispersions of nanocellulose and carbon nanoparticles [39,42]. It should be pointed out that cellulose nanoparticles are excellent dispersive agents for carbon nanoparticles, as they prevent the aggregation of these nanoparticles and maintain them in long-term stable homogeneous suspensions without the need to subject them to chemical functionalization [45,46]. Suspensions of cellulose and carbon nanoparticles are also starting materials for the creation of 3D nanocellulose/nanocarbon composites in the form of aerogels, foams or sponges [45,47,48,49,50]. In addition, composite 3D scaffolds, especially for tissue engineering and for regenerative medicine, can be fabricated by 3D printing using bioinks based on cellulose and carbon nanoparticles [51,52]. Both 2D composites and 3D composites can also be created by adding carbon nanoparticles to cultures of cellulose-producing bacteria, such as *Gluconacetobacter xylinus*. These nanoparticles are then incorporated into bacterial nanocellulose in situ during its growth [53,54,55,56,57]. Another approach is via the electrospinning or wet spinning of solutions containing cellulose and carbon nanoparticles [58,59,60].

Nanocellulose/nanocarbon composites exhibit several more advantageous properties than materials containing only cellulose nanoparticles or only carbon nanoparticles. Adding carbon nanoparticles to nanocellulose materials can further increase their mechanical strength [59,61]. At the same time, the presence of nanocellulose promotes the flexibility and stretchability of the materials [52,62,63]; for a review, see [64]. Adding graphene, carbon nanotubes or boron-doped diamond nanoparticles endows nanocellulose materials with electrical conductivity [39,50,57,65,66]. Other advantageous properties of nanocellulose/nanocarbon composites include their thermal stability [67,68,69], tunable thermal conductivity and optical transparency [48,57,70], intrinsic fluorescence and luminescence [26,71,72] photothermal activity [56], hydrolytic stability [61], nanoporous character and high adsorption capacity [49,61]. Nanocellulose/nanocarbon composites can therefore be used in a wide range of industrial and technological applications, such as water purification [22,29,43,49,54,56,61,73,74,75,76], the isolation and separation of various molecules [22,74,77,78,79], energy generation, storage and conversion [21,23,44,47,64,80,81,82,83,84,85], biocatalysis [86], food packaging [67,68,69,87], construction of fire retardants [48], heat spreaders [70] and shape memory devices [38,88,89,90]. These composites are also used as fillers for various materials, usually polymers, in order to improve their mechanical, electrical and other physical and chemical properties [67,68,69,87,91].

In addition, nanocellulose/nanocarbon composites are promising for biomedical applications, though these applications are less frequent than industrial applications. Biomedical applications include radical scavenging [34,92], photothermal ablation of pathogenic bacteria [93], photodynamic and combined chemophotothermal therapy against cancer [35,94], drug delivery [16,28,65,72,95,96,97], biosensorics [31,32,33,63,66,71,91,98,99,100,101,102,103,104], and particularly tissue engineering and wound dressings. Hybrid materials containing nanocellulose and nanocarbons stimulated the growth and osteogenic differentiation of human bone marrow mesenchymal stem cells [37,59]. They provided good substrates for the attachment, growth and differentiation of SH-SHY5Y human neuroblastoma cells [51] and PC12 neural cells, particularly under electrical stimulation [105]. They enhanced the outgrowth of neurites from rat dorsal root ganglions in vitro and stimulated nerve regeneration in rats in vivo [106]. They also promoted the growth of vascular endothelial cells, enhanced angiogenesis and arteriogenesis in a chick chorioallantoic membrane model [107], and improved cardiac conduction when applied to surgically disrupted myocardium in dogs [52]. In addition, these materials supported the growth of human dermal fibroblasts [108] and mouse subcutaneous L929 fibroblasts [58,62], promoted wound healing in vivo in mice [109] and showed an antibacterial effect [30]. These materials are therefore promising for bone, neural and vascular tissue engineering, for creating cardiac patches and for advanced wound dressings. The biomedical applications of nanocellulose/nanocarbon composites are summarized in Table 1.

This review summarizes recent knowledge on the types, properties and applications of nanocellulose/nanocarbon-based hybrid materials, particularly in biotechnology, biomedicine and tissue engineering, and reports on the experience acquired by our group.

## 2. Nanocellulose/Fullerene Composites

### 2.1. Characterization of Fullerenes

Fullerenes are spheroidal cage-like molecules composed entirely of carbon atoms (Figure 1a). Fullerenes with 60 and 70 carbon atoms (C_60_ and C_70_) are the most stable molecules, and they are therefore most frequently used in industrial and biomedical applications. Fullerenes were discovered in 1985 by Sir Harold Walter Kroto (1939–2016) and his co-workers Richard Smalley, Robert Curl, James Heath and Sean O’Brien. Kroto, Smalley and Curl were awarded the Nobel Prize in 1996. Fullerenes were named after Richard Buckminster “Bucky” Fuller (1895–1983), an American architect, designer, futurist, inventor, poet and visionary, who designed his geodesic dome on similar structural principles (for a review, see [12,13,18,119,120]). Fullerenes are carbon nanoparticles with diverse biological activities. This is due to the fact that they can act as either acceptors or donors of electrons (for a review, see [121]). The acceptor activity can lead to oxidative damage to cell components, such as DNA, cell membrane, mitochondria and various enzymes, to the activation of inflammatory reactions, and to cell apoptosis. These harmful effects of fullerenes can, however, be utilized for photodynamic therapy against tumors and pathogenic microorganisms (for a review, see [122]). The electron donor activity is associated with quenching oxygen radicals, which can be used in protecting skin against UV irradiation, in anti-inflammatory therapy against osteoarthritis, in cardioprotection during ischemia, in neuroprotection during amyloid-related diseases, damage by alcohol or heavy metals, in obesity treatment and in the treatment of diabetes-related disorders. Due to their structural analogy with clathrin-coated vesicles, fullerenes are also promising candidates for drug and gene delivery (for a review, see [12,13,18,119]).

However, fullerenes have low solubility in many solvents, especially in water. This is a major drawback for their wider application in biomedical applications. The water solubility of fullerenes can be achieved by functionalizing them with hydrophilic groups, but this approach does not solve problems arising from the aggregation and clustering of fullerenes. In addition, the formation of singlet oxygen, which is needed for photodynamic therapy, decreases after functionalization due to the perturbation of the fullerene π system. These problems can be mitigated by the complexation of fullerenes with water-soluble agents, including nanocellulose [35].

### 2.2. Preparation and (Bio)Application of Nanocellulose/Fullerene Composites

Cellulose nanocrystals (CNCs) were used to create nanocellulose/fullerene composites. These nanocrystals are typically produced by acid hydrolysis of cellulose fibers, employing either sulfuric acid or hydrochloric acid in order to destroy the amorphous regions of the cellulose, while the crystalline segments remain intact. CNCs can have a needle-like or rod-like morphology, and are also referred to as nanowhiskers or nanorods. This morphology is characterized by a high aspect ratio (i.e., high length to diameter ratio), and thus by a relatively large surface area. In addition, CNCs have a wide range of other advantageous properties, such as high mechanical resistance, broad chemical-modifying capacity, renewability, biodegradability and low cytotoxicity [34,35]; for a review, see [2]. From these points of view, CNCs were considered ideal for immobilization of fullerene nanoparticles [92]. A scheme of preparation of nanocellulose/fullerene composites is depicted in Figure 1b. Composites of CNCs with fullerenes C_60_ were prepared by amine functionalization of CNCs and by subsequently grafting C_60_ onto the surface of amine-terminated CNCs [34]. Conversely, functionalized fullerenes, e.g., polyhydroxylated fullerenes C_60_(OH)_30_, were conjugated with the surface of CNCs [92]. Both of these composites showed a higher radical scavenging capacity in vitro than fullerenes alone, and therefore are promising for biomedical application in antioxidant therapies, e.g., as components of skin care products. In the third type of composites, both cellulose nanocrystals and fullerenes were functionalized, i.e., amino-fullerene C_60_ derivatives were covalently grafted onto the surface of 2, 2, 6, 6-tetramethylpiperidine-1-oxylradical (TEMPO)-oxidized nanocrystalline cellulose [35]. These composites hold promise for photodynamic cancer therapy (Table 1). When these composites were added to the culture medium of human breast cancer MCF-7 cells in the dark, they were taken up by these cells without changes in the cell viability, as revealed by a resazurin assay. However, when irradiated with light, these composites showed dose-dependent toxicity for MCF-7 cells [35].

However, fullerenes are less widely used in nanocellulose/nanocarbon composites than other carbon allotropes, particularly graphene and carbon nanotubes. More frequently, fullerenes are incorporated into a non-nanostructured cellulose matrix. For example, fullerene C_70_, characterized by a strong thermally activated delayed fluorescence at elevated temperatures, which is extremely oxygen sensitive, was incorporated into ethyl cellulose, i.e., a highly oxygen-permeable polymer. This composite was used for construction of an optical dual sensor for oxygen and temperature [115]. An oxygen sensor was constructed using isotopically enriched carbon-13 fullerene C_70_, dissolved in an ethyl cellulose matrix [116]. Mixed-matrix membranes, consisting of ethyl cellulose as a continuous matrix and fullerenes C_60_ as a dispersed phase, were prepared for propylene/propane separation [78]. Electrospun cellulose acetate nanofibers reinforced with fullerenes were used in the construction of dry-type actuators [123]. Cellulose impregnated with fullerenes C_60_ dissolved in o-xylene showed greater extraction efficiency for Cu^2+^, Ni^2+^ and Cd^2+^ ions from an aqueous environment than the pure polymer [124]. Biocompatible composites containing polysaccharides (cellulose, chitosan and gamma-cyclodextrin) and fullerene derivatives (amino-C_60_ and hydroxy-C_60_) were developed for various applications ranging from dressing and treating chronically infected wounds to nonlinear optics, biosensors, and therapeutic agents [118].

## 3. Nanocellulose/Graphene Composites

### 3.1. Characterization of Graphene

Graphene is a single layer of sp^2^-hybridized carbon atoms arranged into a two-dimensional honeycomb-like lattice (Figure 2a). In other words, graphene is a one-atom-thick layer of graphite. It is a basic building block for other carbon allotropes, such as fullerenes, carbon nanotubes and graphite. Graphene is a very thin, nearly transparent sheet, but it is remarkably strong (about 100 times stronger than steel), and highly electrically and thermally conductive (for a review, see [19,20,125,126]). Graphene can be prepared by various methods, which can be divided into two main categories, namely the top-down approach and the bottom-up approach. The top-down approaches include treatment of graphite by mechanical or electrochemical exfoliation, intercalation or sonication, and also nanotube slicing. The bottom-up approaches include growth of graphene from carbon-metal melts, epitaxial growth of graphene on silicon carbide, the dry ice method, and deposition methods such as chemical vapor deposition or dip coating a substrate with graphene oxide (GO), followed by GO reduction (for a review, see [19,20,125,126]). Graphene can be prepared in the form of monolayer or bilayer sheets, nanoplatelets, nanoflakes, nanoribbons and nanoscrolls. Chemically, graphene-based materials include pure graphene sheets, GO or reduced graphene oxide (rGO). Pure graphene sheets can be produced by mechanical exfoliation of graphite or by chemical vapor deposition. GO, a highly oxidative and water-soluble form of graphene, can be obtained by the exfoliation of graphite oxide. Reduced GO can be prepared by chemical, thermal or pressure reduction, and even by bacteria-mediated reduction of GO, which improves its electrical properties (for a review, see [13,18,19,37,125,126]). Graphene and graphene-based materials hold a great promise not only for a wide range of industrial and technology applications, but also for biomedical applications, such as drug, gene and protein delivery, photothermal therapy, construction of biosensors, bioimaging, antimicrobial treatment, and also as scaffolds for tissue engineering (for a review, see [20]).

### 3.2. Preparation and Industrial Application of Nanocellulose/Graphene Composites

Similarly as in fullerenes, cellulose nanoparticles in the form of nanofibrils and nanocrystals increase the dispersion of graphene nanoparticles in water-based environments and prevent their aggregation without the need to subject them to chemical functionalization [46]. A water-based dispersion is the starting material for fabricating nanocellulose/graphene composites (Figure 2b). These composites can be created by filtration [127], filtration combined with hot pressing for fabricating films [128], or by freeze-drying [75] and freeze-casting [48] for fabricating 3D materials, such as aerogels and foams. Other methods are deposition of graphene on a nanocellulose layer [43] and incorporation of graphene into nanocellulose during its biological synthesis by bacteria [54,55,56].

All forms of nanocellulose and graphene have been used for constructing nanocellulose/graphene composites, i.e., CNFs, CNCs, unmodified graphene, GO and rGO. In order to modulate the properties of nanocellulose/graphene composites for specific applications, these materials can be further enriched by various substances, such as metallic or ceramic nanoparticles, oxides, carbides, sulfides, vitamin C, synthetic and natural polymers, enzymes and antibodies. For example, nanocellulose/graphene composites have high adsorption, filtration and photocatalytic ability, and they are therefore widely used for water purification, e.g., for removing antibiotics [75], dyes [43], heavy metals, such as Cu^2+^, Hg^2+^, Ni^2+^ and Ag^+^ [61,76], or for their bactericidal effect [56]. The water-cleaning capacity of these composites can be further enhanced by introducing additional photocatalytic agents, i.e., palladium nanoparticles [54] or zinc oxide (ZnO) nanoparticles [73]. An optimized ultrafiltration membrane for water purification was constructed from polyvinylidene fluoride (PVDF), modified by cellulose nanocrystals functionalized with common bactericides, such as dodecyl dimethyl benzyl ammonium chloride, ZnO and GO nanosheets [129]. Another important additive is vitamin C, which reduces the GO in nanocellulose/GO composites, increases the surface area of the material and increases pore formation, and thus enhances the capacity of the composites for water purification [49]. A combination of rGO-coated cellulose nanofibers with hydrophobic and oleophilic trimethyl chlorosilane enhanced the adsorption capacity of this composite, which is necessary for effective removal of oil-based pollutants from water [74].

Another important industrial application of nanocellulose/graphene composites is in energy storage, generation and conversion. Devices for these purposes include supercapacitors [64,80,130], hydrogen storage devices [131], electrodes for hydrogen evolution reaction [132], lithium ion batteries [133], actuators [81], solar steam generators [82] and electric heating membranes [83]. These devices can be based on pure nanocellulose/nanocarbon composites without additives [80,81,82,83]. However, they often contain additives such as manganese oxide (MnO), which contributes to faradaic pseudocapacitance in supercapacitors [130] or polypyrrole, which acts as an insulator, but its oxidized derivatives are good electrical conductors [134]. Other additives are palladium or platinum nanoparticles for enhanced hydrogen storage [131], nitrogen-doped molybdenum carbide nanobelts in electrocatalysts for hydrogen evolution reaction [132], and silicon oxide nanoparticles in lithium ion batteries [133].

Other important industrial applications of nanocellulose/graphene composites are in the construction of fire retardants, shape memory devices, biocatalysts and materials for food packaging. Super-insulating, fire-retardant, mechanically strong anisotropic foams were produced by freeze-casting suspensions of cellulose nanofibers, GO and sepiolite nanorods, and they performed better than traditional polymer-based insulating materials [48]. Shape memory devices are based on GO/CNC thin films and nanomembranes [38,88] or on GO introduced into a nanocellulose paper made of nanofibers extracted from sisal fibers [89]. An example of a biocatalyst is a nanocellulose/polypyrrole/GO nanocomposite for immobilization of lipase, a versatile hydrolytic enzyme. This biocatalyst was employed for synthesizing ethyl acetoacetate, a fruit flavor compound [86]. Food packaging materials were constructed by filling CNCs and rGO, either separately or in the form of CNC/rGO nanohybrids, into poly (lactic acid) (PLA) matrix or in a poly (3-hydroxybutyrate*-co-*3-hydroxyvalerate) (PHBV) matrix. These composite materials exhibited better mechanical properties than the pristine polymers, and possessed antibacterial activity. In addition, the composites with CNC/rGO nanohybrids performed better than those with a single component nanofiller, i.e., either CNCs or rGO. Due to their antibacterial activity, antioxidant properties and good in vitro cytocompatibility, these composites are also promising for biomedical applications, e.g., as scaffolds for tissue engineering [67,68,69,87].

### 3.3. Biomedical Application of Nanocellulose/Graphene Composites

The biomedical applications of nanocellulose/nanographene composites include photothermal ablation of pathogenic bacteria, combined chemo-photothermal therapy against cancer, drug delivery, biosensorics, isolation and separation of various biomolecules, wound dressing and particularly tissue engineering (Table 1).

For photothermal ablation of pathogenic bacteria, a composite paper containing nanocellulose with Au linked to GO using quaternized carboxymethyl chitosan was developed. When excited by near-infrared laser irradiation, the paper generated a rise in temperature of more than 80 °C, sufficient for photothermal ablation, both on Gram-positive bacteria (*Bacillus subtilis* and *Staphylococcus aureus*) and on Gram-negative bacteria (*Escherichia coli* and *Pseudomonas aeruginosa*). Additionally, the composite paper showed a remarkable enhancement in tensile strength, bursting index and tear index in comparison with the properties of pure nanocellulose paper [93].

For chemophotothermal synergistic therapy of colon cancer cells, dual stimuli-responsive polyelectrolyte nanoparticles were developed by layer-by-layer (LbL) assembly of aminated nanodextran and carboxylated nanocellulose on the surface of chemically modified GO. Tests on the HCT116 human colon cancer cell line revealed that these nanoparticles allowed for the intracellular delivery of curcumin, which was released in response either to acidic environments or to near-infrared excitation [94]. In this context, nanocellulose/graphene composites are good candidates as carriers for controlled drug delivery, particularly of anticancer drugs. Systems releasing doxorubicin, a model drug with broad-spectrum anticancer properties, were developed. These systems included nanocomposite carboxymethyl cellulose/GO hydrogel beads [95], nanocomposite films made of graphene quantum dots incorporated into a carboxymethyl cellulose hydrogel [72] or macroporous polyacrylamide hydrogels. These hydrogels were prepared using an oil-in-water Pickering emulsion, containing GO and hydroxyethyl cellulose with a quaternary ammonium group [96].

Sensing and biosensing is another important application of nanocellulose/graphene composites. These sensors can be divided into electrochemical, piezoelectric, optical and acoustic wave-based sensors. Electrochemical sensors were constructed for detecting cholesterol [98], glucose and pathogenic bacteria [110], avian leucosis virus [111] and organic liquids [112]. The biosensor for detecting cholesterol was based on chemically-modified nanocellulose, grafted with silylated GO and enriched with ZnO nanoparticles in order to enhance its electrical conductivity [98]. The biosensor for detecting glucose and pathogenic bacteria was based on nanocellulose paper coated with GO, reduced by vitamin C and functionalized with platinum nanoparticles with a cauliflower-like morphology in order to enhance the electrical conductivity of the composite. The platinum surface was functionalized either with glucose oxidase (via chitosan encapsulation) or with an RNA aptamer [110]. The biosensor for the avian leucosis virus was an immunosensor, based on graphene-perylene-3, 4, 9, 10-tetracarboxylic acid nanocomposites as carriers for primary antibodies, on composites of nanocellulose and Au nanoparticles as carriers for secondary antibodies, and on the alkaline phosphatase catalytic reaction [111]. The sensor for organic liquids, mainly organic solvents, was based on cellulose nanocrystal-graphene nanohybrids, selectively located in the interstitial space between the natural rubber latex microspheres [112].

Piezoelectric nanocellulose/graphene-based sensors have usually been designed for strain sensing, i.e., as wearable electronics for monitoring the motion of various parts of the human body, e.g., fingers [63,99,113]. For these purposes, the flexibility and stretchability of nanocellulose was further enhanced by adding other polymers, such as elastomers, represented e.g., by polydimethylsiloxane (PDMS) [113], or hydrogels, represented e.g., by poly(vinyl alcohol) (PVA), crosslinked (together with cellulose nanofibers and graphene) with borax [63].

Optical sensors can be based on surface-enhanced Raman spectroscopy (SERS) or fluorescence. Cellulose SERS strips decorated with plasmonic nanoparticles, termed graphene-isolated-Au-nanocrystals (GIANs), were developed for constructing portable sensors for detecting complex biological samples, e.g., for detecting free bilirubin in the blood of newborns [100]. A fluorescence sensor, based on sulfur and nitrogen-co-doped graphene quantum dots, immersed into nanocellulosic hydrogels, was developed for detecting laccase. This enzyme is widely used in industrial and technological applications, such as bleaching of fabrics, tooth whitening, decoloration of hair, water purification and in oxidizing dyes in beer, must and wines [71].

An example of an acoustic wave-based sensor is an ammonia sensor, based on a quartz crystal microbalance (QCM) with a sensing coating. This coating is composed of negatively-charged electrospun cellulose acetate nanofibers, positively-charged polyethylenimine and negatively-charged GO, and it was created by the electrostatic LbL self-assembly technique [101].

For protein isolation, a metal affinity carboxymethyl cellulose-functionalized magnetic graphene was prepared by successive modifications of GO nanosheets with magnetic nanoparticles, carboxymethyl cellulose and iminodiacetic acid, and then chelated with copper ions. This composite exhibited high adsorption selectivity toward histidine-rich proteins, which was utilized for isolating hemoglobin from human whole blood, and also for isolating a polyhistidine-tagged recombinant protein from *Escherichia coli lysate*, namely *Staphylococcus aureus* enterotoxin B [77]. For macromolecular separation, cellulose acetate nanocomposite ultrafiltration membranes were fabricated using 2D layered nanosheets, e.g., GO and exfoliated molybdenum disulfide (MoS_2_), and were successfully tested using macromolecular bovine serum albumin [79].

Nanocellulose/graphene composites are also important components of tissue engineering scaffolds, improving their mechanical properties and their bioactivity. In the studies by Pal et al. (2017), mentioned above, a PLA/CNC/rGO nanocomposite film showed antibacterial activity against Gram-positive *Staphylococcus aureus* and against Gram-negative *Escherichia coli*. At the same time, this film exhibited negligible cytotoxicity against a mouse NIH-3T3 fibroblast cell line, as revealed by an MTT assay of the activity of mitochondrial oxidoreductase enzymes [87]. Nanocomposites of CNCs and rGO, incorporated into PLA matrix through the melt-mixing method, were noncytotoxic and cytocompatible with epithelial human embryonic kidney 293 (HEK293) cells [68]. PLA incorporated with rGO and TEMPO-oxidized CNCs, grafted with poly(ethylene glycol) (PEG), displayed radical scavenging activity and negligible toxicity and cytocompatibility to mouse embryonic C3H10T1/2 cells [69]. A composite film consisting of hydrophilic bacterial cellulose nanofibers and hydrophobic rGO, prepared from GO using a bacterial reduction method, supported the adhesion, viability and proliferation of human bone marrow mesenchymal stem cells in a similar way to standard cell culture polystyrene, and better than pure rGO films [37]. Incorporating GO into electrospun cellulose acetate nanofibrous scaffolds enhanced the adhesion and growth of human umbilical cord mesenchymal stem cells. It also enhanced osteogenic differentiation of these cells, manifested by the activity of alkaline phosphatase, and biomineralization of the scaffolds in a simulated body fluid [59]. Nanofibrous composites of bacterial nanocellulose, a conductive poly(3,4-ethylene dioxythiophene) (PEDOT) polymer and GO, mimicking the native extracellular matrix and allowing electrical stimulation of neural PC12 cells, induced specific orientation and differentiation of these cells [105]. Polyvinyl alcohol/carboxymethyl cellulose (PVA/CMC) scaffolds loaded with rGO nanoparticles, prepared by lyophilization, enhanced the proliferation of EA.hy926 endothelial cells in vitro and angiogenesis in vivo using a chick chorioallantoic membrane model [107]. Polyacrylamide‒sodium carboxymethylcellulose hybrid hydrogels reinforced with GO and/or CNCs also have potential for tissue engineering applications due to their tunable mechanical properties [135]. Genetically modified hydrophobin, a fungal cysteine-rich protein, was used to connect nanofibrillated cellulose of wood origin and graphene flakes in order to construct biomimetic mechanically-resistant materials similar to nacre and combining high toughness, strength and stiffness [136].

Nanocellulose/graphene composites also have great potential for the fabrication of antibacterial textiles and for advanced wound dressing. Antibacterial textiles were prepared by electrospinning a mixture containing cellulose acetate, TiO_2_ and GO sheets. These textiles showed high antibacterial activity with an inhibition rate higher than 95% against *Bacillus subtilis* and *Bacillus cereus* [137]. Bacterial cellulose is considered as one of the most suitable materials for advanced wound dressing, due to its appropriate mechanical properties, such as strength, Young’s modulus, elasticity and conformability, and also due to its great capacity to retain moisture in the wound (for a review, see [2]). These favorable properties can be further enhanced by adding graphene-based materials and by crosslinking with synthetic polymers, such as poly (ethylene glycol), poly (vinyl alcohol), poly (acrylic acid) and poly (acrylamide). In a study by Chen et al. (2019), a bacterial nanocellulose-grafted poly (acrylic acid)/GO composite hydrogel was prepared as a potential wound dressing. The inclusion of GO improved the attachment and proliferation of human dermal fibroblasts in cultures on the composites [108]. Similarly, hydroxypropyl cellulose matrix incorporated with GO and silver-coated ZnO nanoparticles showed improved tensile strength, and also anti-ultraviolet, antibacterial and immunostimulatory effects, which promoted wound healing in an in vivo mouse model [109].

## 4. Nanocellulose/Carbon Nanotube Composites

### 4.1. Characterization of Carbon Nanotubes

Carbon nanotubes (CNTs; Figure 3a) are tubular structures formed by a single cylindrical graphene sheet (single-walled carbon nanotubes, referred to as SWCNTs or SWNTs) or several graphene sheets arranged concentrically (multiwalled carbon nanotubes, referred to as MWCNTs or MWNTs, which also include double-walled CNTs (DWCNTs [16]), and few-walled CNTs (FWCNTs [17]). Carbon nanotubes were discovered as a by-product of fullerene synthesis, and were first described by Iijima et al. (1991) [138]. CNTs have a high aspect ratio (i.e., length to diameter ratio) and thus a relatively large surface area. Their diameter is on the nanometer scale (e.g., from 0.4 nm to 2–3 nm in single-walled nanotubes), but their length can reach several micrometers or even centimeters. Due to these properties, CNTs are suitable candidates for hydrogen storage, for the removal of contaminants from water and air, and also for drug delivery. CNTs have excellent mechanical properties, mainly due to the sp^2^ bonds. The tensile strength of SWCNTs has been reported to be almost 100 times higher than that of steel, while their specific weight is about six times lower. CNTs can therefore be used for reinforcing various synthetic and natural polymers for industrial and biomedical applications, e.g., for hard tissue engineering. When added to a polymer matrix, CNTs can resemble inorganic mineral nanoparticles in the bone tissue, and they can form nanoscale irregularities on the surface of 2D materials and in the pores of 3D materials, which improve the cell adhesion and growth. CNTs are electrically conductive and enable electrical stimulation of cells, which further improves the adhesion, growth and differentiation of cells (for a review, see [10,11,12,13,14,18,139]). However, free CNTs can be cytotoxic, which is attributed to their ability to cause oxidative damage, and also to their contamination with transition metals (e.g., Fe, Ni, Y), which serve as catalysts during CNT preparation. Methods for producing metal-free CNTs have therefore been developed, e.g., arc-discharge evaporation of graphite rods [139].

CNTs also resemble CNFs from the point of view of their morphology and their mechanical properties. For example, highly crystalline, thick CNFs derived from tunicates exhibited mean strength of 3–6 GPa, which was comparable with commercially available MWCNTs. However, the mean strength of other types of CNFs is lower; for example, in wood-derived CNFs the mean strength ranged from 1.6 to 3 GPa [140]. CNTs therefore improve the mechanical strength of nanocellulose/CNT composites, and endow them with electrical conductivity, similarly as graphene. As a result, nanocellulose/CNT composites are used in similar industrial and biomedical applications as nanocellulose/graphene composites, e.g., water purification, energy generation, storage and conversion, filling polymeric materials, constructing sensors and biosensors, drug delivery, cancer treatment, electrical stimulation of tissues, and tissue engineering.

### 4.2. Preparation and Industrial Application of Nanocellulose/CNT Composites

The nanocellulose component in the nanocellulose/CNT composites is used in the form of nanofibrils and nanocrystals, and carbon nanotubes usually in the form of SWCNTs or MWCNTs. Similarly as in composites containing graphene and other carbon allotropes, cellulose nanoparticles facilitate the homogeneous dispersion of CNTs in aqueous environments (Figure 3b), where the two types of nanoparticles are linked by noncovalent interactions, e.g., hydrophobic and electrostatic interactions [42,46]. The dispersion of CNTs can be further facilitated by TEMPO-mediated oxidation of cellulose nanoparticles, which endows them with abundant anionic carboxyl groups [141]. Other ways include the functionalization of CNTs with self-assembling amphiphilic glycosylated proteins [142] or the use of oil-in-water Pickering emulsions of cellulose nanoparticles [143]. From the aqueous dispersions, 2D and 3D nanocellulose/CNT composites can be formed, e.g., by vacuum filtration, centrifugal cast molding, foam forming, casting and printing [39,45,50,141]. Pickering emulsions of cellulose nanocrystals and SWCNTs or MWCNTs were used for fabricating aerogels and foams by freeze-drying [143]. Similarly as in nanocellulose/graphene composites, CNTs can be added to cultures of bacteria producing nanocellulose, and incorporated into the bacterial nanocellulose during its growth [53,57]. Composite nanocellulose/SWCNT films can be transparent or semitransparent [39,45,141], and can transmit radiant energy [144].

Like nanocellulose/graphene composites, nanocellulose/CNT composites can be combined with various atoms, molecules and nanoparticles in order to enhance their properties for specific applications. For example, Ag nanoparticles attached to the surface of MWCNTs influenced the electrochemical properties of CNT-based films developed on a bacterial nanocellulose membrane [36]. Nanocellulose and CNTs can be also used as additives to various hybrid materials. For example, a hybrid material, created by combination of poly (3,4-ethylenedioxythiophene)-poly (styrenesulfonate) (PEDOT:PSS), silver nanoparticles (AgNPs), CNTs and a nanocellulose layer, was used for constructing a tactile sensor [103]. Incorporation of polypyrrole-coated CNTs into chemically cross-linked CNC aerogels created promising materials for flexible 3D supercapacitors [145]. A combination of cellulose acetate, chitosan and SWCNTs with Fe_3_O_4_ and TiO_2_ in electrospun nanofibers enables combined removal of Cr^6+^, As^5+^, methylene blue and Congo red from aqueous solutions via the adsorption and photocatalytic reduction processes [146].

Energy-related applications of nanocellulose/CNT composites include biofuel cells, varactors, supercapacitors, and electrodes for lithium batteries, thermoelectric generators for heat-to-electricity conversion or for constructing heating elements [50]. A biofuel cell comprising electrodes based on supercapacitive materials, i.e., on CNTs and a nanocellulose/polypyrrole composite, was utilized to power an oxygen biosensor. Laccase, immobilized on naphthylated MWCNTs, and fructose dehydrogenase, adsorbed on a porous polypyrrole matrix, were used as cathode and anode bioelectrocatalysts, respectively [84]. Another biofuel cell was based on a conductive MWCNT network, developed on a bacterial nanocellulose film, and functionalized with redox enzymes, including pyroquinoline quinone glucose dehydrogenase (anodic catalyst) and bilirubin oxidase (cathodic catalyst). This system generated electrical power via the oxidation of glucose and the reduction of molecular oxygen [44]. Microelectromechanical system varactors, i.e., voltage-controlled capacitors, consisted of a freestanding SWCNT film, which was employed as a movable component, and a flexible nanocellulose aerogel filling [85]. Supercapacitors with high physical flexibility, desirable electrochemical properties and excellent mechanical integrity were realized by rationally exploiting the unique properties of bacterial nanocellulose, CNTs, and ionic liquid-based polymer gel electrolytes [147]. Other flexible 3D supercapacitor devices were fabricated by incorporating polypyrrole nanofibers, polypyrrole-coated CNTs, and manganese dioxide (MnO_2_) nanoparticles in chemically cross-linked cellulose nanocrystal aerogels [145]. Electrospun core-shell nanofibrous membranes, containing CNTs stabilized with cellulose nanocrystals, were developed for use as high-performance flexible supercapacitor electrodes with enhanced water resistance, thermal stability and mechanical toughness [40]. Electrodes for lithium batteries were based on freestanding LiCoO_2_/MWCNT/cellulose nanofibril composites, fabricated by a vacuum filtration technique [148], or on freestanding CNT‒nanocrystalline cellulose composite films [41]. Thermoelectric generators for heat-to-electricity conversion were based on large-area bacterial nanocellulose films with an embedded/dispersed CNT percolation network, incorporated into the films during nanocellulose production by bacteria in culture [57]. Electrical energy can also be converted into thermal energy, as demonstrated by the composite structure of wood-derived nanocellulose, MWCNTs and pulp, designed for a heating element application [50].

### 4.3. Biomedical Application of Nanocellulose/CNT Composites

Biomedical applications of composites of cellulose and CNTs are summarized in Table 1. These composites are important systems for drug delivery. The CNTs in the composites can be conjugated with many therapeutics, usually anticancer drugs, but also other types of drugs. For example, an osmotic pump tablet system coated with cellulose acetate membrane containing MWCNTs was developed for delivery of indomethacin (for a review, see [16,65]). Cancer cells can be killed by nanocellulose‒CNT dispersions even without the presence of anticancer drugs. For example, nonmercerized type-II cellulose nanocrystals in dispersions with SWCNTs displayed cytotoxicity for human epithelial colorectal adenocarcinoma Caco-2 cells, but they enhanced the mitochondrial metabolism of normal cells [149].

Construction of sensors and biosensors is another important application of nanocellulose/CNT composites useful for (bio) technology and medicine. A tactile piezoresistance and thermoelectric-based sensor, mentioned above, which is capable of simultaneously sensing temperature and pressure, is fabricated from TEMPO-oxidized cellulose, PEDOT:PSS, AgNPs and CNTs [103]. Another pressure sensor was developed using aerogels consisting of plant cellulose nanofibers and functionalized few-walled CNTs [17]. Highly conductive and flexible membranes with a semi-interpenetrating network structure, fabricated from MWCNTs and cellulose nanofibers, showed the electrical features of capacitive pressure sensors and were promising for various electronics applications, e.g., touch screens [150]. Advanced flexible strain sensors for controlling the human body motion were fabricated by pumping hybrid fillers consisting of CNTs/CNCs into porous electrospun thermoplastic polyurethane membranes [91]. Other strain sensors were fabricated by a facile latex assembly approach, in which CNCs played a key role in tailoring the percolating network of conductive natural rubber/CNT composites [114]. A water-responsive shape memory hybrid polymer, based on a thermoplastic polyurethane matrix crosslinked with hydroxyethyl cotton cellulose nanofibers and MWCNTs, was also developed for constructing a strain sensor [90]. A flexible and highly sensitive humidity sensor, capable of monitoring human breath, was based on TEMPO-oxidized nanofibrillated cellulose and CNTs [104]. An electrochemical biosensor for three adenosine triphosphate (ATP) metabolites, namely uric acid, xanthine and hypoxanthine, was based on a composite of NH_2_-MWCNT/black phosphorene/AgNPs, dispersed in carboxymethyl cellulose [102]. Another electrochemical molecularly-imprinted sensor was based on a nanofibrous membrane prepared by the electrospinning technique from cellulose acetate, MWCNTs and polyvinylpyrrolidone, and was used for determining ascorbic acid [151]. An oxygen biosensor powered by a biofuel cell containing MWNCTs, a nanocellulose/polypyrrole composite, laccase and fructose dehydrogenase, was mentioned above [84]. Versatile wearable textile sensors, e.g., for gas sensing, were produced from cellulose nanofibers extracted from tunicates, homogeneously composited with SWCNTs, by wet spinning in an aligned direction [60].

Other important biomedical applications of nanocellulose/CNT composites are in electrical stimulation of cells and tissues in order to improve their regeneration and function, and in tissue engineering. For example, stretchable, flexible and electrically conductive biopatches for restorating conduction in damaged cardiac regions and for preventing arrhythmias were prepared. These patches were based on nanofibrillated cellulose/SWCNT ink three-dimensionally printed onto bacterial nanocellulose. They restored cardiac conduction after its disruption by a surgical incision made in the ventricular part of the heart in experimental dogs [52]. For neural tissue stimulation, multiblock conductive nerve scaffolds with self-powered electrical stimulation were prepared. These scaffolds were based on polypyrrole/bacterial nanocellulose composites with platinum nanoparticles on the anode side for glucose oxidation, and nitrogen-doped CNTs on the cathode side for oxygen reduction. These scaffolds enhanced the elongation of neurites outgrowing from rat dorsal root ganglions in vitro and stimulated nerve regeneration in a rat sciatic nerve gap model in vivo in comparison with composites containing only polypyrrole and bacterial nanocellulose. These scaffolds could replace the metal needles that are currently used for external electrical stimulation of neural tissue, which may cause pain and a risk of infection [106]. 3D printing was also used for creating scaffolds based on a conductive ink composed of wood-derived CNFs and SWCNTs. These scaffolds were intended for neural tissue engineering for experimental brain studies, and supported the attachment, growth and viability of human neuroblastoma SH-SHY5Y cells [51]. Other scaffolds for tissue engineering consisted of electrospun cellulose acetate nanofibers, assembled with positively-charged chitosan and negatively-charged MWCNTs via an LbL technique. These scaffolds promoted the adsorption of cell adhesion-mediating molecules from the serum supplement of the culture medium and the adhesion and growth of mouse subcutaneous L929 fibroblasts [117]. Our own results related to the potential application of nanocellulose/CNT composites as scaffolds for tissue engineering are reported in Appendix A.

## 5. Nanocellulose/Nanodiamond Composites

### 5.1. Characterization of Nanodiamond

Diamond is an allotrope of carbon, consisting of carbon atoms arranged in a cubic crystal structure covalently bonded in sp^3^ hybridization (Figure 4a). Like all nanostructured materials, nanodiamonds or diamond nanoparticles are defined as features not exceeding 100 nm in at least one dimension, although some larger diamond particles, i.e., 125–210 nm, are still referred to as nanodiamonds (for a review, see [152]). At the same time, the size of ultrananocrystalline diamond particles is 3–5 nm [153,154]. Diamond nanoparticles can be prepared by various methods. The most widely used techniques are detonation of carbon-containing explosives in an oxygen-deficit environment and microwave-enhanced plasma chemical vapor deposition (MECVD). Other techniques include the radiofrequency plasma-assisted chemical vapor deposition (PACVD) method, milling of diamond microcrystals, hydrothermal synthesis, ion bombardment, laser bombardment, ultrasound synthesis and electrochemical synthesis (for a review, see [10,152,153,154,155,156]). Nanodiamonds are considered to be the most advanced carbon materials in the world. This is due to their excellent mechanical, optical, electrical, thermal and chemical properties. The mechanical properties of nanodiamonds include the highest hardness of all materials on earth, a high Young’s modulus, high fracture toughness, high pressure resistance and a low friction coefficient. Their optical properties include transparency, high optical dispersion, and their ability to display various colors and to emit intrinsic luminescence (fluorescence), which is due to defects in the diamond lattice or contamination of the lattice with foreign atoms, such as N, B, H, Ni, Co, Cr or Si. Regarding their electrical properties, nanodiamonds can act as good insulators in their pristine state and as semiconductors after doping, usually with boron. Their thermal properties include superior thermal conductivity and low thermal expansion. The chemical properties of nanodiamonds include low chemical reactivity and resistance to liquid- and gas-phase oxidations. However, nanodiamonds can be doped with various atoms, and their surface can be functionalized by various atoms, chemical groups and (bio)molecules ([152,153,157,158]; for a review, see [10,11,12,13,14,15,155]).

### 5.2. Preparation and (Bio)Application of Nanocellulose/Nanodiamond Composites

There is greater use of diamond nanoparticles than of fullerenes in nanocellulose/nanocarbon composites, but diamond nanoparticles are used less than graphene and carbon nanotubes. This may be because a nanodiamond is more expensive and is electrically nonconductive in its pristine state. In composites with nanofibrillated cellulose (Figure 4b), a nanodiamond was used for constructing highly thermally conductive, mechanically resistant and optically transparent films with potential application as lateral heat spreaders for portable electronic equipment [70]. A highly mechanically resistant and optically transparent nanopaper was made of cationic CNFs and anionic nanodiamond particles by filtration from a hydrocolloid and subsequent drying [159]. Moreover, a diamond can be rendered electrically semiconductive by doping it with boron, and then can be used for constructing biosensors. For example, a sensor for biotin was developed by the adsorption of captavidin, a nitrated avidin with moderate affinity to biotin, on a carboxymethylcellulose layer stabilized on a boron-doped diamond electrode by a Nafion film. This biosensor was used for analyzing biotin in blood plasma [66] (Table 1).

The reinforcing effect of diamond nanoparticles, coupled with their optical transparency, has also been used advantageously for other biomedical applications, particularly for wound dressing. Nanocellulose/nanodiamond composites are more mechanically resistant than purely nanocellulose-based materials, but they retain their flexibility and stretchability. In addition, their optical transparency enables direct inspection of wounds without the need to remove the dressing. For example, incorporating diamond nanoparticles in a concentration of 2 wt % into chitosan/bacterial nanocellulose composite films resulted in a 3.5-fold increase in the elastic modulus of these films. These composite films were transparent, but their transparency can be modulated by the concentration of diamond nanoparticles, turning them gray and semitransparent at higher nanodiamond concentrations (3 and 4 wt %). The viability of mouse subcutaneous L929 fibroblasts in cultures on these films, evaluated by an MTT test of the activity of cell mitochondrial enzymes, was more than 90% at 24 h after seeding. However, at 48 h, it had dropped to about 75%, which indicated that diamond nanoparticles are slightly cytotoxic [62]. A similar result was obtained on L929 fibroblasts grown on electrospun composite nanofibrous mats containing chitosan, bacterial cellulose and 1–3 wt % medical-grade nanodiamonds [58]. The viability of these cells, estimated by the MTT assay, dropped from approx. 90% on day 1 to approx. 75% on day 3. Nevertheless, the addition of nanodiamonds facilitated the electrospinning process, reduced the diameter of the nanofibers in the mats, regulated the water vapor permeability of the mats, enhanced their hydrophilicity and improved their mechanical properties to a similar level as in native skin [58].

Adding diamond nanoparticles per se did not significantly increase the antibacterial activity of chitosan/bacterial nanocellulose composites [62]. This activity can be further enhanced, e.g., by adding silver nanoparticles [160]. Nanocellulose/nanodiamond composites can also act as a suitable platform for drug delivery. This was demonstrated on transparent doxorubicin-loaded carboxylated nanodiamonds/cellulose nanocomposite membranes, which are promising candidates for wound dressings. These membranes are porous, transparent, with appropriate mechanical properties, and without doxorubicin they are noncytotoxic for HeLa cells [97].

## 6. Composites of Nanocellulose with Other Carbon Nanoparticles

In addition to fullerenes, graphene, nanotubes and nanodiamonds, other important carbon nanoparticles used in industrial, biotechnological and biomedical applications include carbon nanofibers, carbon quantum dots and nanostructures formed by activated carbon and carbon black. All these nanomaterials can be used in nanocellulose/nanocarbon composites. The biomedical applications of these composites are summarized in Table 1.

### 6.1. Composites of Nanocellulose and Carbon Nanofibers

Carbon nanofibers can be created by carbonization of cellulose nanofibers originating from bacterial nanocellulose [21,23,161], urea [161], filter paper [162] or plant-derived cellulose [22,24]. Another method of preparing carbon nanofibers is chemical vapor deposition (CVD; [25]). These carbon nanofibers can be further combined with other carbon nanoparticles, mainly graphene. For example, a composite paper consisting of nitrogen-doped carbon nanofibers, reduced graphene oxide (rGO) and bacterial cellulose was designed as a high-performance, mechanically tough, and bendable electrode for a supercapacitor. The bacterial nanocellulose in this paper is exploited both as a biomass precursor for the creation of carbon nanofibers by pyrolysis and as a supporting substrate for the newly-created material [21]. In another study, highly conductive freestanding cross-linked carbon nanofibers, derived from bacterial cellulose in a rapid plasma pyrolysis process, were used as substrates for the growth of vertically-oriented graphene sheets for constructing alternating current filtering supercapacitors [23]. A small amount of rGO can also act as an effective initiator of carbonization of cellulose nanofibers through microwave treatment [24]. Carbonization of aerogels, prepared from a mixture of PVA, cellulose nanofibers and GO by freeze-drying, enhanced the hydrophobic properties, the specific surface area and the adsorption capacity of these aerogels. These materials then became suitable candidates for oil‒water separation and environmental protection [22]. In addition to graphene, cellulose-derived carbon nanofibers can be combined with various other nanoparticles and nanostructures, such as Pt nanoparticles for methanol oxidation reaction [161], TiO_2_ films and Fe_3_O_4_ nanoparticles for lithium ion batteries [162], tin oxide (SnO) nanoparticles for lithium‒sulfur batteries [163] or NiCo_2_S_4_ nanoparticles for hydrogen evolution reaction [164]. Carbon nanofibers are also promising for biomedical applications, particularly bone tissue engineering. Their nanoscale diameter produced a nanoscale surface roughness of their compacts or of their composite with poly-lactic-*co*-glycolic acid (PLGA). This nanoroughness promoted preferential adhesion of osteoblasts from other cell types, particularly fibroblasts, which could prevent fibrous encapsulation of bone implants [25].

### 6.2. Composites of Nanocellulose and Carbon Quantum Dots

Carbon quantum dots (CQDs) are quasispherical carbon nanoparticles (less than 10 nm in diameter) with a chemical structure and physical properties similar to those of graphene oxide. These nanoparticles emit a strong wavelength-dependent fluorescence. By changing the CQD size, the color of the emitted light can be tuned from deep ultraviolet to visible and near-infrared light. In addition, the fluorescence of CQDs, and also their water solubility, can be further modulated by functionalizing their surface with various atoms, chemical functional groups and molecules, such as metals, carboxyl groups, organic dyes and polymers. CQDs present good photostability, low photobleaching and relatively low cytotoxicity, and they are therefore considered to be suitable for biomedical applications such as bioimaging, biosensing, photodynamic and photothermal therapy of cancer, and drug delivery [165].

In hybrid materials with nanocellulose, CQDs were applied for constructing biosensors and drug delivery systems, and also for water purification. An optical sensor for visual discrimination of biothiols was based on a bacterial cellulose nanopaper substrate with ratiometric fluorescent sensing elements. These elements included N-acetyl l-cysteine capped green cadmium telluride (CdTe) quantum dots‒rhodamine B and red CdTe quantum dots‒carbon dots [26]. Hybrid materials containing carbon quantum dots and cellulose are also promising carriers for drug delivery. Composite core/shell chitosan-poly (ethylene oxide)-carbon quantum dots/carboxymethyl cellulose-poly(vinyl alcohol) nanofibers were prepared through coaxial electrospinning as a biodegradable implant for local delivery of temozolomide (TMZ), an anticancer drug. When tested in vitro, the antitumor activity of TMZ conjugated with carbon quantum dots against the tumor U251 cell lines was higher than the activity of the free drug [28]. Last but not least, carbon quantum dots, homogeneously dispersed together with magnetic Fe_3_O_4_ nanoparticles in electrospun cellulose nanofibers, were promising for the removal of Hg(II) ions from water [27].

### 6.3. Composites of Nanocellulose and Activated Carbon

Activated carbon is a form of carbon processed to have small, low-volume pores that increase the surface area, which is then available for the adsorption and removal of various toxic contaminants and microorganisms. Composite membranes consisting of a bilayer of porous activated carbon and TEMPO-oxidized plant-derived CNFs showed high capability for removing *Escherichia coli* from water [29]. Activated carbon was also a component of a wound dressing material consisting of a polyvinyl alcohol and cellulose acetate phthalate polymeric composite film, reinforced with Cu/Zn bimetal-dispersed activated carbon micro/nanofibers. This material suppressed the growth of *Pseudomonas aeruginosa*, the most prevalent bacteria in infected wounds caused by burns, surgery and traumatic injuries [30].

### 6.4. Composites of Nanocellulose and Carbon Black

Carbon black is a form of paracrystalline carbon, produced industrially by partial combustion or thermal decomposition of gaseous or liquid hydrocarbons under controlled conditions. Carbon black has a high surface-area-to-volume ratio, though not so high as that of activated carbon. Although it is considered to have low toxicity, the International Agency for Research on Cancer has classified it as possibly carcinogenic to humans. In addition, as a component of environmental pollution, carbon black can cause oxidative damage and an inflammatory reaction, which further mediate genotoxicity, reproductive toxicity, neurotoxicity and diseases of the respiratory and cardiovascular systems [166,167]. Nevertheless, carbon black is currently used as a filler in tires and in other rubber products, and as a pigment in inks, paints and plastics.

Composites of nanocellulose and carbon black have been used mainly for constructing biosensors, particularly wearable sensors for strain and human body motion, e.g., motion of the fingers, the elbow joint and the throat. A strain-sensing device with excellent waterproof, self-cleaning and anticorrosion properties was based on a superhydrophobic electrically conductive paper. This paper fabricated by dip-coating a printing paper into a carbon black/carbon nanotube/methyl cellulose suspension and into a hydrophobic fumed silica suspension [33]. Another strain-sensing device was fabricated by printing carbon black conductive nanostructures on cellulose acetate paper. At the same time, this material had electrochemical properties promising for the detection of hydrogen peroxide [31]. An electrochemical aptasensor for detecting *Staphylococcus aureus*, e.g., in human blood serum, was designed as a nanocomposite of Au nanoparticles, carbon black nanoparticles and cellulose nanofibers, and was endowed with a thiolated specific *S. aureus* aptamer as a sensing element [32].

## 7. Potential Cytotoxicity and Immunogenicity of Nanocellulose/Nanocarbon Composites

The vast majority of studies dealing with potential biomedical applications of nanocellulose/nanocarbon composites have reported no cytotoxicity or negligible cytotoxicity of these composites, namely of nanocellulose/fullerene composites [35], nanocellulose/graphene composites [68,69,72,87,107], nanocellulose/CNT composites [51,149] and nanocellulose/nanodiamond composites [58,62,97]. Composites containing other carbon nanoparticles, such as activated carbon nanoparticles [30] or carbon quantum dots [28] have also shown no significant cytotoxic effects. In the mentioned studies, cytotoxicity was mainly tested in vitro on various cell types, such as fibroblasts, epithelial and endothelial cells, and mainly on cell lines, including tumor cell lines. The cell viability and proliferation were usually evaluated by tests of the activity of mitochondrial enzymes, such as MTT assay, resazurin (Alamar Blue) assay, or by a direct microscopic examination of the cells. Some composites have also been tested in vivo, e.g., in a rat model (nanocellulose/CNT composites; [106]), a canine model (nanocellulose/CNT composites; [52]), or using a chick chorioallantoic membrane model (nanocellulose/graphene composites, [107]), without adverse effects.

However, the individual components of nanocellulose/nanocarbon composites, particularly carbon nanoparticles, can act as cytotoxic, if they are not bound to any matrix and are free to move. Graphene and graphene-based carbon nanomaterials, such as fullerenes and nanotubes, are hydrophobic in their pristine state, and can enter into hydrophobic interactions with cholesterol in the cell membrane, which can be extracted from the membrane. In this manner, carbon nanoparticles can damage cells even without penetrating them. Another mechanism of cell membrane damage is the generation of reactive oxygen species (ROS) by carbon nanoparticles. In addition, the nanoparticles can penetrate the cell membrane, and can cause oxidative damage to mitochondria, and can also enter the cell nucleus and act as genotoxic agents (for a review, see [10,12,19,20]). Nanodiamonds have been considered to be relatively nontoxic in comparison with other carbon nanoparticles. However, as shown in our earlier studies, hydrophobic, hydrogen-terminated and positively-charged diamond nanoparticles can enter the cells, impair their growth and cause cell death [152,156]. The mechanism of cell damage by nanodiamonds is by generating ROS, and by excessive delivery of sodium ions adsorbed on the nanodiamond surface [168]. Last but not least, carbon nanoparticles can be immunogenic, i.e., they can activate inflammatory reactions, which can be, as has been demonstrated on carbon black, the main pathogenic mechanism of respiratory, cardiovascular and other serious diseases [166,167].

Cellulose nanoparticles, which are generally considered to be biocompatible [34,114] and of a low ecological toxicity [169], can also act as cytotoxic and immunogenic. It has even been speculated that, due to their high aspect ratio and stiffness, CNCs may cause similar pulmonary toxicity as carbon nanotubes and asbestos [170]. In a mouse model, cellulose nanocrystals induced oxidative stress, caused pulmonary inflammation and damage, increased levels of collagen and transforming growth factor-beta (TGF-β) in lungs, and impaired pulmonary functions [170]. In addition, these effects were markedly more pronounced in female mice than in male mice. The immunogenicity of CNCs was also proven in vitro. CNCs and their cationic derivatives CNC-aminoethylmethacrylate and CNC-aminoethylmethacrylamide evoked an inflammatory response in mouse macrophage J774A.1 cells and in peripheral blood mononuclear cells by increasing the level of ROS in mitochondria, the release of ATP from mitochondria and by stimulating the secretion of interleukin-1beta (IL-1β) [171]. The cytotoxicity and immunogenicity of CNCs depend on the preparation conditions and are increased under harsh and caustic conditions, e.g., the so-called mercerization process, i.e., an alkali treatment [149]. CNFs can also cause cytotoxicity and oxidative damage, which can be even more pronounced than in the case of CNCs, and can evoke an inflammatory response (for a review, see [2,172]). The potential cytotoxicity and immunogenicity of nanocellulose, nanocarbon and their composites should therefore be taken into account when they are for use in biomedical applications.

## 8. Conclusions

Nanocellulose/nanocarbon composites and other hybrid materials containing cellulose nanoparticles (nanofibrils or nanocrystals) and carbon nanoparticles (fullerenes, graphene, carbon nanotubes, nanodiamonds and other carbon nanoparticles) are novel materials that are promising for a wide range of applications in industry, (bio)technology and medicine. This is due to their unique properties, such as high mechanical strength coupled with flexibility and stretchability (composites with graphene, carbon nanotubes and nanodiamond), shape memory (composites with graphene and carbon nanotubes), photodynamic and photothermal activity (composites with fullerenes and graphene), electrical conductivity (composites with graphene and carbon nanotubes), semiconductivity (composites with boron-doped diamond), thermal conductivity (composites with graphene and nanodiamonds), tunable optical transparency (composites with single-walled carbon nanotubes and nanodiamonds), intrinsic fluorescence and luminescence (composites with graphene quantum dots and carbon quantum dots), and high adsorption and filtration capacity (composites with graphene, carbon nanotubes and carbon quantum dots). These properties arise mainly from the advantageous combination of nanocellulose and nanocarbon, which associates and enhances the desirable effects of each of these components. These materials can be prepared relatively easily from a water-based suspension, which is advantageous particularly for biomedical applications. These applications include drug delivery, biosensorics, isolation of various biomolecules, electrical stimulation of damaged tissues, and particularly tissue engineering (bone, neural and vascular) and wound dressing. Our results have proven supportive effects of nanocellulose/carbon nanotube composites on the adhesion and growth of human and porcine adipose tissue-derived stem cells, particularly under dynamic cultivation in a pressure-generating lab-made bioreactor (see Appendix A). However, it should be pointed out that the biomedical applications of nanocellulose/nanocarbon composites are associated with the risk of their potential cytotoxicity and immunogenicity, although this risk appears to be lower than for the single components of these materials.

## Figures and Tables

**Figure 1 nanomaterials-10-00196-f001:**
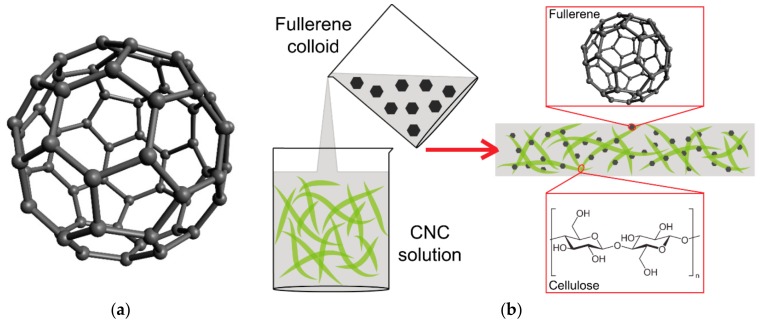
Scheme of fullerene C_60_ (**a**) and of the preparation and structure of nanocellulose/fullerene composites (**b**).

**Figure 2 nanomaterials-10-00196-f002:**
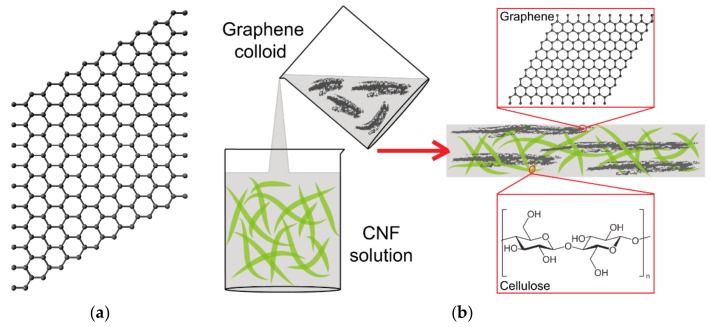
Scheme of graphene (**a**) and of the preparation and structure of nanocellulose/graphene composites (**b**).

**Figure 3 nanomaterials-10-00196-f003:**
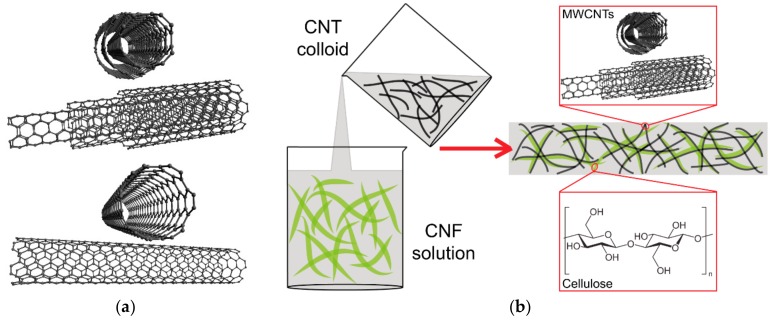
Scheme of multi-walled and single-walled carbon nanotubes (**a**) and of the preparation and structure of nanocellulose/carbon nanotube composites (**b**).

**Figure 4 nanomaterials-10-00196-f004:**
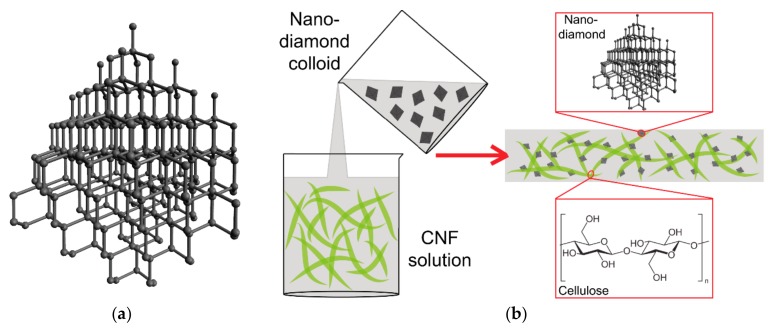
Scheme of a nanodiamond (**a**) and of the preparation and structure of nanocellulose/nanodiamond composites (**b**).

**Table 1 nanomaterials-10-00196-t001:** Biomedical applications of nanocellulose/nanocarbon composites.

Application	Nanocellulose/Nanocarbon Composites Containing:				
	Fullerenes	Graphene	CNTs	Nanodiamonds	Others
**Radical scavenging**	NH_2_-CNC/C_60_ [34]; CNC/C_60_(OH)_30_ [92]				
**Photodynamic cancer therapy**	TEMPO-oxidized CNC/C_60_-NH_2_ [35]				
**Photothermal, chemo-photothermal therapy**		Bacteria: [93]			
		Cancer: [94]			
**Drug delivery**		Anticancer drugs (doxorubicin) [72,95,96]	Anticancer and other drugs [16]	Anticancer drugs (doxorubicin) [97]	Carbon quantum dots: Anticancer drugs (temozolomide) [28]
**(Bio)sensors**		Electrochemical: cholesterol [98]; glucose and bacteria [110]; avian leucosis virus [111]; organic liquids [112]	Electrochemical: ATP metabolites [102]; oxygen [84]	Electrochemical: Biotin [66]	Carbon black: Electrochemical aptasensor for *S. aureus* [32]; electrochemical sensor for H_2_O_2_ [33]
		Piezoelectric: strain, human motion [63,99,113]	Piezoresistance and thermoelectric-based: pressure and temperature [103]; pressure [17]; strain, human motion [90,91,114]; humidity, human breath [104]		Carbon black: Strain, human motion [31,33]
	Optical: oxygen and temperature [115]; oxygen [116]	Optical: *SERS*: bilirubin [100]; *Fluorescence*: laccase [71]			Carbon quantum dots: optical sensor for biothiols [26]
		Acoustic: ammonia [101]			
**Isolation of biomolecules**		Histidine-rich proteins, hemoglobin [77]; bovine serum albumin [79]			
**Electrical stimulation of tissues**			Cardiac tissue [52]; neural tissue [106]		
**Tissue engineering (TE)**		General cell biocompatibility [68,69,87]; bone TE [37,59]; neural TE [105]; vascular TE [107]	Neural tissue engineering [51]; TE in general [117]		
**Wound dressing/healing**	Polysaccharides/fullerene C_60_ derivatives [118]	Human dermal fibroblasts in vitro [108]; mouse model in vivo [109]		L929 fibroblasts in vitro [58,62]; HeLa cells in vitro, wound dressings delivering doxorubicin [97]	Activated carbon: antibacterial wound dressing [30]

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
