# Peer review of "Applications of Nanocellulose/Nanocarbon Composites: Focus on Biotechnology and Medicine"

_nanomaterials, 2020, doi:10.3390/nano10020196_

Round 1
Reviewer 1 Report
See the file below : COMMENTS .docx

Author Response
Line 65 : …nanodiamonds ( for a review see 10-20 ) . The references seem related only to nanodiamonds
Answer:
These references are not related only to nanodiamonds, but they cover homogeneously all types of carbon nanoparticles, at least those “classical” - fullerenes, graphene-based particles, carbon nanotubes and nanodiamonds. The references 10 and 11 are book chapters dealing with fullerenes, carbon nanotubes and nanodiamonds. The reference 12 is a book chapter dealing purely with fullerenes. The reference 13 is a review article dealing with fullerenes, graphene, carbon nanotubes and nanodiamonds. The reference 15 is a book chapter on nanostructures materials in general, including polymeric, metallic, ceramic and carbon nanoparticles (fullerenes, graphene, carbon nanotubes, nanodiamonds). Only the reference 15 is a book chapter dealing purely with nanodiamonds. The reference 16 is a review article focused on carbon nanotubes and their potential in drug delivery. The reference 17 is a communication article on nanocellulose/CNT composites. The reference 18 is a review article on photodynamic therapy, where fullerenes, graphene and carbon nanotubes are described and characterized. The references 19 and 20 are review articles on graphene and related materials and their potential biomedical applications.
Line 129 A fullerene molecule can contain from 20 to several hundreds or even thousands of carbon atoms. NOOOOOO!!!!
Answer:
This statement was based on a paper by Janner A. Alternative approaches to onion-like icosahedral fullerenes. Acta Crystallogr A Found Adv., 2014, 70, 168-180, where fullerenes with 240, 540, 960, 1500, 2160 and more carbon atoms are mentioned. Another source of this statement was a monograph of David Tomanek “Guide through the Nanocarbon Jungle: Buckyballs, Nanotubes, Graphene, and Beyond”, published by Morgan & Claypool, IOP Science eBooks, 2014 (ISBN-13: 978-1627052726 ISBN-10: 1627052720), especially the supplementary material to this book. This material contains a pictorial table of fullerenes, where the fullerene C20 is considered as the smallest possible fullerene, and the other fullerenes are drawn up to the fullerene C720. Nevertheless, it is true that the most stable fullerenes are C60 and C70, while the fullerene with too small or too high number of carbon atoms are unstable and existing rather theoretically (i.e. designed and optimized by computation) than practically. Therefore, the sentence “A fullerene molecule can contain from 20 to several hundreds or even thousands of carbon atoms” was deleted from the manuscript.
Line 138 … they can act either as acceptors or as donors of electrons. At this point the support of a reference is needed . I suggest to cite this fundamental book , written by an outstanding scientist : E.F. Sheka “ Fullerenes : Nanochemistry, Nanomagnetism, Nanomedicine , Nanophotonics “ CRC Press Taylor and Francis Group , 2011, Chapter 1
Answer:
The reference of the chapter 1 of the mentioned book has been added (reference 112).
Line 142 … photodynamic terapy . Chapter 9 of the same book
Answer:
The reference of the chapter 9 of the mentioned book has been added (reference 112).
Line 199 …single layer of sp2- and sp3-hybridized carbon atoms . NOOOO !!!! Carbon in graphene is only sp2-hybridized
Answer:
This statement was adopted from a review article by Jin et al. 2015 (ref. 18) but it is generally known that graphene is typically sp2-hybridized. The mention on sp3 hybridization was deleted from the manuscript.
Line 203 It is necessary to insert a reference on the properties of graphene , of GO and of rGO.
Answer:
The references describing the properties of graphene, GO and rGO are mentioned in the line 211, together with the preparation of these carbon nanoparticles. However, for better clarity of the text, they have also been added to the text describing the properties of the mentioned nanocarbons (lines 200-204).
Line 211 nano-onions, fibers, quantum dots ..3D structures, such as foams or scaffolds.. All these forms are usually considered graphitic nanocarbons, not graphene
Answer:
The mentioned form of nanocarbon were adopted mainly from the reference 13. We have deleted these forms from the sentence, and we have reformulated it according to the reviewer’s suggestion: “Graphene can be prepared in the form of monolayer or bilayer sheets, nanoplatelets, nanoflakes, nanoribbons, and nanoscrolls”.
Line 254 These electronic devices . Some of the cited devices are not “ electronic “
Answer:
We have reformulated the sentence according to the reviewer´s suggestion and we have left the word “electronic” out of the text.
Line 392 … or several graphene sheets arranged concentrically (multi-wall carbon nanotubes, referred to as MWCNTs or MWNTs). In addition, double-wall CNTs [16] and few-wall CNTs [17] were also described . Several , double-wall and few-wall CNT are all concentrical arrangement of graphene sheets . Better divide CNT simply in SW (one) and MW ( two, few, many)
Answer:
Originally, we have described the basic division of CNTs on SWCNTs and MWCNTs, and we also referred to some papers where some other „special“ types of CNTs were mentioned. Now we have modified the sentence according to the reviewer´s suggestion:
“Carbon nanotubes (CNTs) are tubular structures formed by a single cylindrically-shaped graphene sheet (single-wall carbon nanotubes, referred to as SWCNTs or SWNTs) or several graphene sheets arranged concentrically (multi-wall carbon nanotubes, referred to as MWCNTs or MWNTs, which also include double-wall CNTs (DWCNTs [16]), and few-wall CNTs (FWCNTs [17])”.
Line 397 ,,,relative to their molecular weight . Attention, CNT are not molecules, they have not a molecular weight
Answer:
The formulation “molecular weight” was originally adopted from the review article by Yin et al. 2015 (reference 18). According the reviewer suggestion, the sentence as reformulated as follows: CNTs have a high aspect ratio (i.e. length to diameter ratio) and thus a relatively large surface area.
Line 428 .. linked by non-covalent bonds, e.g. hydrophobic and electrostatic interactions . Attention, it seems that hydrophobic interactions are a type of bonding
Answer:
To the best of our knowledge, the hydrophobic interactions (or the hydrophobic effect) are not covalent bonding, i.e., electron-sharing bonding. We have modified the sentence as “….linked by non-covalent interactions, e.g. hydrophobic and electrostatic interactions”.
Line 676 ..good adhesiveness to the underlying substrate, and good interlayer cohesion . These are not general properties of the nanosized diamond, but are related only to nanodiamond films/layers/coatings grown on some substrates ( the adhesion of CVD diamond and the interlayer cohesion depend on the material of the substrate)
Answer:
Good adhesiveness to the underlying substrate, and good interlayer cohesion are properties which were adopted from a book chapter by Bacakova et al. 2016 (reference 15). However, as mentioned by the referee, these properties are related to nanodiamond films deposited on certain substrates. Therefore, these properties were deleted from the text.
Line 683 -resistance to wet etching. Better to detail: oxidations? But diamonds are very resistant also to gas-phase oxidations.
Answer:
The sentence has been modified as “The chemical properties of nanodiamonds include low chemical reactivity and resistance to liquid- and gas-phase oxidations”.
Line 720. A general reflection . The discussion of data reported in Ref 62 can be extended also to other example of nanocomposites reported in the present review .In many of the papers cited in this review , the active material 3 is a complex mixture of several components, where nanocellulose/nanocarbons systems represent only one of the components . One wonders if the performances exhibited by such materials are really due to the nanocellulose/nanocarbon composites or rather to the other components : Fe3O4 ( Ref. 27 ) , Ag ( Ref. 36 and 62 ) , PVA/BORAX ( Ref.63) , Pt (Ref.127 and 159 ) , Au (Ref. 93, 128) , TiO2 (Ref .133 ) , Fe3O4 and TiO2 (Ref.142 and 160 ) , NiCo2S4 (Ref. 162) and so on .
Answer:
The performances are generally due to the nanocellulose/nanocarbon composites, but they are markedly enhanced by addition of various components mentioned in our article. At the same time, the desired performances of the additional components are improved by nanocellulose/nanocarbon composites. In other words, the nanocellulose/nanocarbon composites and the additional component act synergistically in order to obtain a desired performance. If either nanocellulose/nanocarbon composites or additional components are absent, the performance is worse than in the hybrid material containing both types of components.
Line 819-830 … have reported no cytotoxicity or negligible cytotoxicity …. components of nanocellulose/nanocarbon composites, particularly carbon nanoparticles, can act as cytotoxic . Here there are discrepancies, please check the text .
Answer:
On the one hand, the nanocellulose/ nanocarbon composites have been generally reported in the scientific literature to be non-toxic or only of a low, negligible cytotoxicity. On the other hand, the cytotoxicity of individual components of these composites, i.e. nanocellulose and particularly nanocarbon components is higher than the whole composite. In other words, the nanocellulose/nanocarbon components are less harmful that their individual components. This apparent discrepancy has been better clarified in the text.
Line 875 - thermal conductivity (composites with graphene and nanodiamonds), thermal insulation (composites with graphene ) . Conductivity or Insulation ?
Answer: In most cases, graphene can introduce thermal conductivity to the composite. However, in some cases, the appropriate addition of nanocellulose can bring insulative properties when needed - in these cases, the nanocarbon benefit comes from other reasons than its thermal conductive properties. For example, Wicklein et al. 2015 (reference 48) describes super-insulating, fire-retardant, mechanically strong anisotropic foams, produced by freeze-casting suspensions of cellulose nanofibers, graphene oxide and sepiolite nanorods, which performed better than traditional insulating materials, based on synthetic polymers, such as expanded polystyrene and polyurethane. In this case, the insulating effect can be attributed mainly to cellulose and sepiolite. However, graphene oxide can also contribute to the insulating effect, because its thermal conductivity can be tailored by tuning its oxidation degree during preparation process. A study by Meng et al. (2018) reported that the cross-plane thermal conductivity of bulk graphene oxide exhibits more than 100 times decrease in comparison with its precursor graphite at room temperature. Nevertheless, in literature, graphene and related materials have been generally reported as materials with a high thermal conductivity, and therefore the mention of thermal insulation was deleted from the text in order to avoid confusion.
Meng Q-L, Liu H, Huang Z, Kong S Jiang P, Bao X. Tailoring thermal conductivity of bulk graphene oxide by tuning the oxidation degree. Chinese Chemical Letters, 2018; 29(5): 711-715, doi: 10.1016/j.cclet.2017.10.028.
Line 886 – Results of a research that is better to eliminate from the text of the review and insert, if the Editor agree , in an added appendix.
Answer: We think that these results are an important contribution to the research on potential biomedical applications of nanocellulose/nanocarbon composites, which are still less developed than the industrial applications. However, we agree to the reviewer that these result inserted in the central part of the review article can disrupt the flow of the main text and can cause some confusion. Therefore, we have moved our research part to the Appendix section of the article.
…… the English form needs a final refinement . My suggestion to the authors is to cut some very long sentences . The shortening of sentences longer more than 2-3 printed lines ( see for example those in lines 161-165 , 523-527, 696-699, 805-808 ) makes more easily the understanding of the text .
Answer: The manuscript was revised by Mr. Robin Healey, a native English speaker, professional linguist, translator and English teacher, who is mentioned in the Acknowledgement section. The long sentences mentioned by the referee, and also other long sentences, have been shortened, i.e. they have usually been divided into two independent sentences. The other long sentences, which have been modified, are now in the lines 85-88, 91-94, 112-116, 187-190, 308-312, 334-337, 357-361, 369-373, 448-452, 654-658, 679-682 and 788-791.
Please, also see the attachment.

Reviewer 2 Report
This paper deals with a review of the research on the applications of medical and biotechnology to composites prepare of nanocellulose and various carbon nanoparticles. The carbon-based nanoparticles reviewed in this paper are well-reviewed for protein cytotoxicity and immunogenicity of composites prepared using fullerene, graphene, carbon nanotube, nanodiamond, etc.
However, it would be better to insert the reference by referring to the schematic diagram or illustration that can express the structural characteristics of each carbon nanoparticle. In addition, if there is no problem of copyright rather than characteristics expressed under certain conditions, it is better to quote the experimental schematic and results in the figure.
Author Response
This paper deals with a review of the research on the applications of medical and biotechnology to composites prepare of nanocellulose and various carbon nanoparticles. The carbon-based nanoparticles reviewed in this paper are well-reviewed for protein cytotoxicity and immunogenicity of composites prepared using fullerene, graphene, carbon nanotube, nanodiamond, etc.
However, it would be better to insert the reference by referring to the schematic diagram or illustration that can express the structural characteristics of each carbon nanoparticle. In addition, if there is no problem of copyright rather than characteristics expressed under certain conditions, it is better to quote the experimental schematic and results in the figure.
Answer:
We have drawn illustration expressing the structural characteristics of each carbon nanoparticle, and also the principle of preparation and scheme of each nanocellulose/nanocarbon composite.

Round 2
Reviewer 1 Report
The authors made the modifications required to improve the text
Reviewer 2 Report
This article has been revised well and I think this article may be published.